# Kinesthetic illusions induced by muscle tendon vibration: The orientation of the vibration motor as a new methodological factor?

Lydiane Lauzier[1]◉, Jacques Abboud[2], François Nougarou[3]◉, Louis-David Beaulieu[1]◉*

**1** Laboratoire BioNR, Centre Intersectoriel en Santé Durable, Université du Québec à Chicoutimi, Chicoutimi, Quebec, Canada, **2** Groupe de Recherche sur les Affections Neuromusculosquelettiques, Département des Sciences de L'Activité Physique, Université du Québec à Trois-Rivières, Trois-Rivières, Quebec, Canada, **3** Laboratoire de Signaux et Systèmes Intégrés, Département de Génie Électrique et Informatique, Université du Québec à Trois-Rivières, Trois-Rivières, Quebec, Canada

◉ These authors contributed equally to this work.
* louis-david_beaulieu@uqac.ca

## Abstract

### Purpose/aim

To investigate the impact of changing the rotational orientation of the vibrating motor on kinesthestic illusions.

### Materials and methods

Twenty healthy individuals received vibration over the wrist flexor muscles of dominant and non-dominant sides (80 Hz, 1 mm, 10 seconds) using four conditions (3 trials/conditions) defined by the rotational direction of the vibrator's eccentric rotating mass according to the anatomical position: (1) proximodistal, (2) distoproximal, (3) mediolateral and (4) lateromedial. Non-parametric statistical analyses were used to compare illusion characteristics across conditions.

### Results

Lateromedial rotation created illusions that were more often in unexpected directions compared to the other rotational orientations. Distoproximal rotation was more likely to evoke kinesthetic illusions of wrist extension (76.8%) compared to lateromedial rotation (57.7%; p = 0.009). The latter led more frequently to complex/combined movement illusions (26.1%) especially with an ulnar deviation component (17.7%) compared to the other rotational directions.

### Conclusion

Results from the present study demonstrated that the rotational orientation can influence illusory perceptions, but not to a great extent. Distoproximal rotation was

**Data availability statement:** All relevant data are within the manuscript and its Supporting information files.

**Funding:** This study was support by Ordre Professionnel de la Physiothérapie du Québec (to L. Lauzier: https://oppq.qc.ca), Réseau Intersectoriel en Santé de l'Université du Québec (to J. Abboud, F. Nougarou and L-D. Beaulieu: https://risuq.uquebec.ca), FRQ Fonds de Recherche du Québec – Santé (FRQS) under Grand number 297854 (to L-D. Beaulieu: https://frq.gouv.qc.ca/sante/) and Réseau Provincial de Recherche en Adaptation-Réadapation (REPAR) (to L. Lauzier: https://repar.ca). The funders had no role in study design, data collection and analysis, decision to publish, or preparation of the manuscript.

**Competing interests:** The authors have declared that no competing interests exist.

more effective to elicit the expected illusions of wrist extension, compared to the lateromedial orientation that more often caused complex and variable perceptions of movement. Distoproximal rotation should thus be preferred if clear and reproducible perceptions are required and lateromedial might serve as a way of creating illusions more akin to everyday functional movements. Although the exact underlying mechanisms remain unclear, our work raises awareness on the importance of gaining a better understanding and control over methodological factors that could affect kinesthetic illusions.

## Introduction

Proprioception refers to the internal sense of movement and positioning of the body and limbs [1]. Proprioceptive deficits are highly prevalent following a stroke or other neurological diseases and interfere with motor control and functional recovery [1,2]. However, their clinical management is often neglected by clinicians due to lack of valid and reliable tools specific to proprioceptive functions [3].

Recently, muscle tendon vibration was proposed as an innovative approach to evaluate and treat proprioceptive disorders [4]. The mechanical vibration of superficial tendons strongly depolarize primary afferents of muscle spindles, therefore sending a proprioceptive message coherent with the stretching of the vibrated muscle [5,6]. In the absence of visual feedback, muscle tendon vibration elicits clear and reproducible illusions of movement in healthy individuals [7,8]. A few studies assessed the clinical utility of kinesthetic illusions in the context of sensorimotor disorders [9–12]. In stroke, vibration-induced illusions were found as less clear on the paretic limb compared to the non-paretic limb and healthy counterparts [9,10], supporting its validity to identify proprioceptive deficits. Also, a repeated application of vibration showed promise as a proprioceptive-based intervention to improve upper-limb sensorimotor control [11,12].

The best vibration parameters to create kinesthetic illusions have been investigated in the past, frequency being the most extensively studied so far. Microneurographic recordings of spindles activity shown their highest firing rate around 80 Hz [5], and stronger illusions are habitually perceived with 60–100 Hz, 80 Hz being the most often selected [4,13,14]. Vibration amplitude received less attention than frequency, but ≥0.2–0.5 mm is generally required to elicit perceptible illusions [5,15,16] and augmenting the amplitude (for example between 1–2 mm) tend to increase the vibration's effects [4,5,13,14,17,18]. Joint positioning and muscle state are also key influencing factors: a slight stretching and complete relaxation of the vibrated muscle group are required to ensure an effective transmission of mechanical displacements from the tendon to muscle spindles [5,15]. A step-by-step procedure and rating sheet, named Standardized Kinesthetic Illusion Procedure, was recently proposed to control these parameters of influence and standardize measurements of illusory perceptions [19].

However, information about how the vibration motor should be placed over the tendon is mostly lacking. Vibration motors come with different sizes, shapes, and

mechanisms to create vibrations. Many of those commercially available are using a rotating system named eccentric rotating mass on which a small weight is mounted off-center from the rotational axis [20]. The centripetal force of the unbalanced mass is asymmetrical, resulting in a net centrifugal force that causes the motor to move, thereby generating vibration [20]. The combination of speed of rotation with the distance and weight of the eccentric mass overall results in the observed vibration frequency and amplitude [21]. For maximal energy transfer to the underlying tendon, the system must be placed so that the internal mass rotates in a way that makes the vibrator moves up and down perpendicularly to the skin. However, since the mass moves in a circular way, the vibrator also oscillates back and forth parallel to the skin and tendons [20]. The overall vibratory behavior can thus be represented as two sinusoidal waves with an amplitude, frequency, and phase that are propagating along the two axes (i.e., perpendicular and parallel to the tendon) [21,22]. From mathematical models and real-time measurements of these two waves, a vibratory orbit can be constructed. Orbit analysis is extensively used in industry to diagnose abnormal vibratory behaviour of rotating machinery [21,22], but has never been considered as a potential methodological factor to control in muscle tendon vibration. Yet, orbit analysis showed that although the eccentric mass rotates in a circular pattern, the resulting movement of the whole vibrator is not circular [21,22]. The vibrator's orbit rather takes an elliptical shape which has a tilted orientation that shifts when changing the rotational direction of the mass (that is, clockwise and counterclockwise) [21,22]. A tilted elliptical orbit could potentially result in an asymetrical distribution of forces transmitted by the vibrator; however evidence from industrial orbit analysis is insufficient to infer if and how this factor could affect kinesthetic perceptions.

Altogether, changing the vibration's position and the rotational direction of the eccentric mass could change the location and direction of the forces transmitted to the targeted tendons. To date, no study verified whether the rotational direction and placement of the vibrator influence or not the perceived illusions – these methodological choices remain arbitrary. Considering the potential of vibration-induced illusions for diagnostic and therapeutic purposes, all potential influencing factors should be rigorously investigated to ensure that a low sensory perception or limited response to treatment are ascribed to meaningful clinical interpretations and not methodological issues.

The main objective of this study was to explore the impact of changing the orientation and rotational direction of the vibrating motor on the perceived kinesthetic illusions in healthy individuals.

## Materials and methods

### Participants

Twenty healthy participants took part in this experiment realized at the BioNR research laboratory (*Université du Québec à Chicoutimi, UQAC*). Based on means, standard deviations and effect sizes from a study having tested the impact of changing vibration's frequency on kinesthetic perceptions [23], the lowest sample size to reach 95% power and an error of 5% was nine participants (calculated with G*Power 3.1.9 software). However, 20 participants were recruited to prevent potential effect size overestimation. Selection criteria were to be aged between 18 and 35 years old and have no neurological or musculoskeletal disorders affecting the upper limb. They had never experienced kinesthetic illusions before and remained naïve to the study's hypothesis. Participants' characteristics are detailed in Table 1. Ethical approbation was obtained before recruitment by *Comité d'éthique de la Recherche de l'UQAC* (#2020−409) and all participants gave their written consent before the beginning of the experiment.

### Experimental procedure

The experiment consisted of one session lasting about 2 hours. First, participants completed 3 questionnaires: (i) a questionnaire about personal characteristics (i.e., age, sex, height, weight, comorbidity, medical background), (ii) the Global Physical Activity Questionnaire (QPAQ) [24] and (iii) the Edinburgh Handedness Inventory short form (EHI) [25]. Clinical measurements were taken bilaterally and included sensitivity to pressure and vibration, respectively assessed using Semmes-Weinstein monofilaments (hand set) [26] and a 128 Hz tuning fork [27]. Wrist proprioception was also tested on

**Table 1. Participant's characteristics and clinical measures.**

| Characteristics | |
|---|---|
| **Sample size (males/females)** | 20 (7/13) |
| **Age (years)** | |
| Mean±SD | 24±3 |
| Range | 18-31 |
| **Weight (lbs)** | |
| Mean±SD | 150.55±26.64 |
| **Height (cm)** | |
| Mean±SD | 170.22±11.75 |
| **Lateral dominance (right/left)** | 16/4 |
| **Edinburgh Handedness Inventory** | |
| Mean±SD | 61.87±66.82 |
| Range | −87.5-100 |
| **Physical activity level (min/week)** | |
| Mean±SD | 2244.80±2298.61 |
| Range | 240-10416 |
| *Clinical measures* | Mean±SD |
| **Monofilament test (grams)** | |
| Dominant | 3.26±0.46 |
| Non-dominant | 3.29±0.46 |
| **Cutaneous vibration sense (s)** | |
| Dominant | 8.01±1.65 |
| Non-dominant | 7.80±1.88 |
| **Bone vibration sense (s)** | |
| Dominant | 8.69±1.99 |
| Non-dominant | 8.96±2.24 |
| **Nottingham assessment sensory scale (sensory subscale/3)** | |
| Dominant | 3.00±0 |
| Non-dominant | 3.00±0 |

SD = standard deviation.

both sides by the *kinesthetic sensations* section of the Nottingham Sensory Assessment [28]. Then, kinesthetic illusions were induced by a custom-made vibratory device. The participant was seated in a standard chair with their forearm supported on a table (shoulder and elbow slightly flexed about 45 degrees). The wrist was supported by the experimenter to control its angular position and to ensure an optimal relaxation throughout the experiment. An elastic strap was installed at the level of the metacarpal heads to reduce tactile feedback from the experimenter holding the hand. A small flexible goniometer was taped on the skin, secured under the elastic strap and aligned with standard anatomic landmarks (i.e., parallel with radius bone and with second metacarpal bone) to keep track of the wrist's angular position (Fig 1). The device consists of a vibration motor (Precision Microdrives, London, UK) with an eccentric mass of 1 g located 1.5 mm off-center encapsulated in a 3D printed plastic box and controlled via MATLAB software (Mathworks, Natick, Massachusetts, USA). The vibrator was strapped over wrist flexor tendons as shown in Fig 1. Vibration parameters, measured by an embedded measurement system on the motor, were a frequency of 80 Hz, 1 mm amplitude and 10 seconds duration [5,8]. The Standardized Kinesthetic Illusion Procedure (SKIP) was strictly followed, which ensured standardization of the directives given to the participant, the identification of an optimal joint position to elicit the clearest illusions and a constant control of

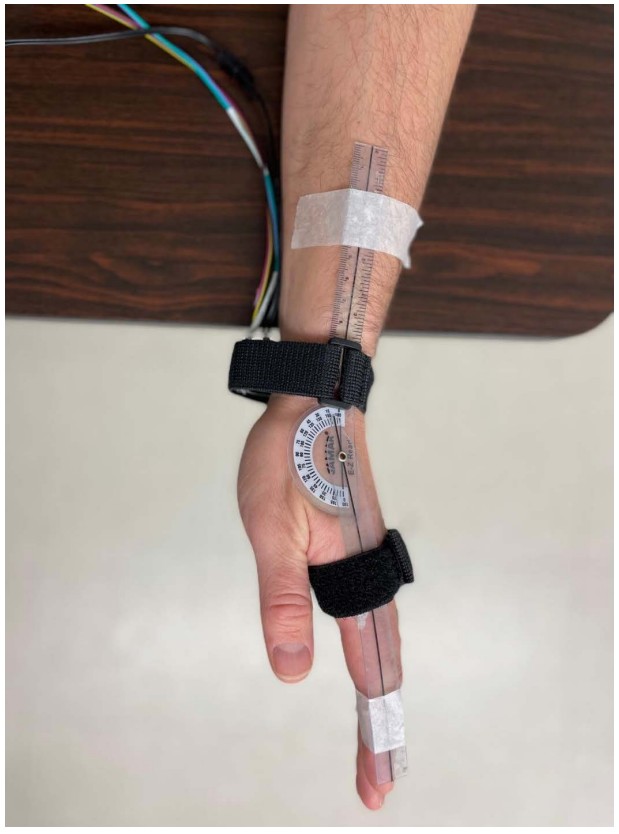

**Fig 1. Positioning of the participant and the vibration device.** The participant was seated with their forearm supported on a table. A goniometer was used to identify the optimal illusion angle (SKIP procedure). The vibration device was strapped and applied over the distal tendons of wrist flexor muscles. The motor's position was validated by palpation of the musculotendinous structures.

the subject's relaxation and sensory feedbacks caused by the physical contact between the experimenter and the subjects (for more details [19]). Prior to the experiment, a few trials were tested with eyes open to ensure setup comfort and habituation to cutaneous vibration sensation. Standardized instructions were given to the participant: "*The test is designed to evaluate how well you can perceive movements at the joint during the vibration. Beyond the sensation of vibration, you will have to tell whether a movement occurred or not and if yes, to give specific details on the perceived movement*". The first step of the SKIP was the determination of an optimal joint position to perceive the clearest illusion. Vibration was first applied at an arbitrary angle sufficient to slightly stretch the targeted tendon. The evaluator then tested different wrist joint angles in the sagittal plane (flexion/extension) using five degrees increments. The angle at which the clearest illusions are perceived was noted by the experimenter using the goniometer strapped on the wrist and used throughout the rest of the experiment. The next step was to assess the illusion clearness and direction at the optimal joint position. Three consecutive trials were realized and for each trial, the subject rated the illusion's clearness and precision (perfectly clear and precise = 3; moderately clear and precise = 2; vague and not precise = 1; no illusion = 0) and direction (illusion in the expected direction (i.e., which would stretch the vibrated tendons) = 1; any other direction = 0). In addition, a visual analogue scale (VAS) was used to evaluate the perceived speed and amplitude of the illusions, where scores near 0 relates to a very low speed and amplitude and scores toward 10 would refer to a very high speed and amplitude [29]. In addition to rate the direction score using the SKIP grading system, the evaluator asked participants to qualitatively describe the perceived direction and/or to reproduce what they felt with their wrist. To quantify and compare the subjective perceptions

of movement reported by the participants, movements were attributed by the evaluator to one of twelve possible perceptions at the wrist (extension, extension with ulnar deviation, ulnar deviation, flexion with ulnar deviation, flexion, flexion with radial deviation, radial deviation, extension with radial deviation, supination, pronation, circumduction, no movement perceived). This information was used to determine the frequency of occurrence for the different illusion directions. SKIP measures were obtained for both the dominant and non-dominant hands in four conditions consisting of different rotational orientations of the vibrating motor, presented in a random order between participants (Fig 2). These directions were defined by how the rotating mass of the motor would first hit the tendons according to the standard anatomical position: (i) proximodistal, (ii) distoproximal, (iii) mediolateral and (iv) lateromedial (see Fig 3). Of note, the vibrator's position was exactly the same for the proximodistal and distoproximal conditions, as it was the same for mediolateral and lateromedial conditions – the only difference was the clockwise and counterclockwise rotational direction of the eccentric mass. Three trials per condition were realized, for a grand total of 480 trials for the whole sample. For SKIP scores of clearness and precision and direction, only the score most often reported by the participant was retained [19]. For VAS scores of speed and amplitude, the mean was calculated from the three trials.

## Data analysis

Statistical analysis was done using SPSS version 26 (Armonk, NY, United States) with an alpha risk below 0.05. Data normality was verified with the Shapiro-Wilk test. Since most of the data did not respect a normal distribution, non-parametric tests were used. Pair-wise comparisons with the Wilcoxon signed-rank test compared SKIP scores of illusion clearness and VAS scores of speed and amplitude between the four conditions of rotational direction. The McNemar test was applied to compare SKIP scores of direction since it is a dichotomous variable. Then, frequency of occurrence (in %) for each movement perceived were calculated and compared across conditions according to the qualitative description of the participants' perceptions. Combined movements were calculated based on the sum of each frequency where we observed a combination of 2 movements (e.g., extension and ulnar deviation, flexion and radial deviation) and complex movements (e.g., supination, pronation and circumduction). The same method was used to obtain totals within a specific component (e.g., all perceptions that included an extension component). The Wilcoxon test was applied on frequency data for pairwise comparisons between conditions. Of note, data from dominant and non-dominant sides were pooled to reach higher

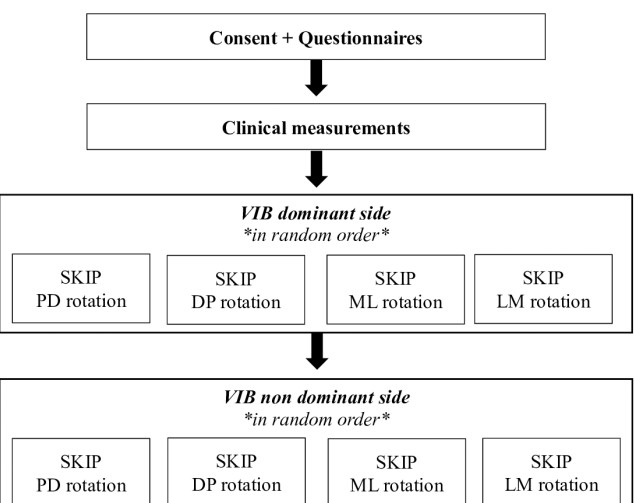

**Fig 2. Experimental procedure.** VIB: muscle tendon vibration; SKIP: Standardized Kinesthetic Illusion Procedure; PD: proximodistal; DP: distoproximal; ML: mediolateral; LM: lateromedial.

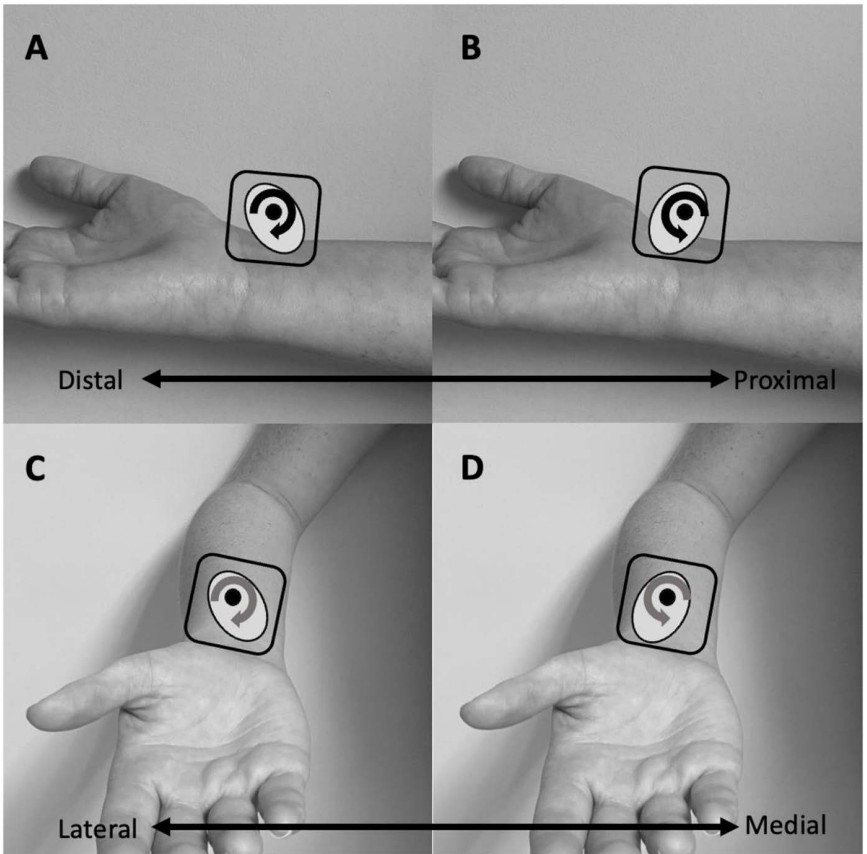

**Fig 3. Experimental conditions of vibration's rotational orientations.** A = Proximodistal rotation; B = Distoproximal rotation; C = Mediolateral rotation and D = Lateromedial rotation.

statistical power. However, secondary analyses also compared illusions between sides with the Wilcoxon test to further explore the potential influence of handedness on sensory perception [30]. The same comparisons were made between sides for clinical measures of sensitivity to pressure and vibration. Bonferroni correction was performed to adjust the level of statistical significance to limit the risk of type I errors (alpha level of 0.05 divided by 6 paired comparisons between rotational conditions = $p \le 0.008$; or kept at $p < 0.05$ for comparisons between dominant and non-dominant sides). Of note, the minimal dataset required to reproduce our analyses is made fully available (cf. supporting information).

## Results

### Effects of rotational orientation on kinesthetic illusions

As shown in Table 2, SKIP clearness scores and VAS scores of speed and amplitude were not significantly different for pooled and non-pooled data between the four conditions of rotational direction (all $p > 0.05$). Clearness of kinesthetic illusions was stable across the four conditions, with participants mostly rating "*moderately clear and precise*" movements. Direction scores showed significantly lower scores (i.e., any other direction than the expected wrist extension) for the lateromedial versus the proximodistal ($p = 0.0002$) and distoproximal conditions ($p = 0.008$). VAS data also pointed toward low to moderate sensations of speed and amplitude, with median scores between 2.58–4.53 and 25th–75th quartiles ranging between 1.37–6.52.

Frequency of occurrence for the 12 possible perceptions of wrist movements reported by the participants were calculated and listed in Table 3. Rate of success for eliciting a kinesthetic illusion, independently of its direction, was high (95.4–100%) for the four conditions of rotational orientation. The most frequent movement perceived by the participants across the four conditions was wrist extension, ranging between 57.7–76.8% of trials. This frequency of wrist extension occurrence was similar between conditions, except for one pair-wise comparison resulting in a significantly lowest frequency for lateromedial compared to distoproximal orientations (p = 0.009). Combined or complex movement illusions were reported in about 18% of all trials, with the highest occurrence for the lateromedial orientation (26.1%). Lateromedial rotation caused illusory movements with an ulnar deviation component in 17.7% of trials, which seemed higher than proximodistal (6.3% - p = 0.049), distoproximal (7.2% - p = 0.016) and mediolateral (5.4% - p = 0.021) conditions, but without reaching the significance level after applying the Bonferroni correction. Conversely, results with the lateromedial condition showed that illusions involving an ulnar deviation component were significantly more frequent than illusions with a radial deviation component (p = 0.004). No other difference was found between conditions and directions of illusory perceptions.

## Effects of manual dominance

Comparisons between dominant and non-dominant sides showed a significant difference for clearness score (p = 0.021) and for the VAS speed and amplitude (p = 0.011), but only when the motor rotated in mediolateral direction (Table 2). In this case, participants felt less clear and lower speed and amplitude of movement illusions when the vibration was applied to the non-dominant side. No difference was observed between sides for direction scores as well as for the other conditions of rotational orientation.

## Discussion

The present study verified the impact of changing the motor's position and rotational direction on vibration-induced kinesthetic illusions. Our results confirmed that these methodological aspects of tendon vibration do influence illusory perceptions in healthy individuals, but not to a great extent. Wrist extension illusions remained the most frequently reported illusion for all conditions of rotational orientation, as expected when targeting wrist flexors [5,7,8]. However, other movements than uniplanar wrist extension were reported for 23.20–42.30% of trials depending on the tested condition, especially when using the lateromedial orientation. These results are discussed below to identify possible explanatory mechanisms and underline their relevance for future research and clinical applications of vibration-induced kinesthetic illusions.

**Table 2. Kinesthetic illusions scores according to the motor's rotational orientation.**

| | | Proximodistal rotation [Median (IQR)] | Distoproximal rotation [Median (IQR)] | Mediolateral rotation [Median (IQR)] | Lateromedial rotation [Median (IQR)] |
|---|---|---|---|---|---|
| **Clearness score SKIP (/3)** | **Pooled** | 2.00 (1.00-2.00) | 2.00 (1.00-2.75) | 2.00 (2.00-3.00) | 2.00 (2.00-3.00) |
| | D | 2.00 (1.00-2.75) | 2.00 (1.25-3.00) | 2.00 (2.00–3.00)[c] | 2.00 (1.00-2.75) |
| | ND | 2.00 (1.00-2.00) | 2.00 (1.00-2.00) | 2.00 (2.00–2.00)[c] | 2.00 (2.00-3.00) |
| **Direction score SKIP (/1)** | **Pooled** | 1.00 (1.00–1.00)[a] | 1.00 (1.00–1.00)[b] | 1.00 (0.25-1.00) | 1.00 (0.00–1.00)[a,b] |
| | D | 1.00 (1.00-1.00) | 1.00 (1.00-1.00) | 1.00 (0.25-1.00) | 0.00 (0.00-1.00) |
| | ND | 1.00 (1.00-1.00) | 1.00 (1.00-1.00) | 1.00 (0.25-1.00) | 1.00 (0.00-1.00) |
| **VAS speed/amplitude (/10)** | **Pooled** | 3.70 (2.31-5.43) | 4.00 (1.98-5.18) | 3.43 (2.17-5.00) | 4.30 (2.91-5.83) |
| | D | 3.65 (2.06-5.55) | 3.87 (1.98-5.62) | 4.03 (3.01–5.53)[c] | 4.22 (2.18-5.49) |
| | ND | 3.93 (2.39-5.40) | 4.00 (1.85-5.05) | 2.58 (1.37–4.34)[c] | 4.53 (3.57-6.32) |

[a,b]Denotes significant difference between conditions of rotational direction after Bonferroni correction (i.e., p ≤ 0.008).

[c]Denotes significant difference between dominant (D) and non-dominant (ND) sides (p < 0.05).

IQR = interquartile range; VAS = 10 points visual analogue scale.

The high success rates and limited contrasts found between the four conditions underscore that even if the position and orientation of an eccentric rotating mass vibrator stays uncontrolled, a relatively high efficacy for inducing kinesthetic illusions can still be expected. However, if a specific and reproducible direction of illusion is desired, for example a wrist extension, then our study stresses out the importance of better controlling the vibrator's position and rotational orientation. The lateromedial condition was less effective in producing the expected illusions of wrist extension, especially compared to the distoproximal orientation. Also, participants reported more frequently movements that included an ulnar component when using the lateromedial orientation. We propose technological and anatomical explanations for these findings. We fixed our vibrating motor in a 3D-printed plastic cover that fit the shape of the motor. This resulted in a surface of contact with the participant having a slight rectangular shape of 1.8 × 2.5 cm. So, flipping the vibrator's orientation by 90 degrees between the distoproximal (or proximodistal) and the lateromedial (or mediolateral) conditions slightly changed the surface of contact over the tendons. The larger 2.5 cm side of the system when using distoproximal/proximodistal position could have better covered all tendons from wrist flexors compared to the smaller 1.8 cm side used for the lateromedial/mediolateral conditions. Although the 0.7 cm difference of length between the two sides might seem negligible, as supported by the similar efficacy for inducing illusions between conditions, our results still encourage testing further the impact of changing the shapes and sizes of vibrators, like a 2.5 × 2.5 cm surface of contact. However, the particular directions of illusions reported with the lateromedial condition cannot be solely explained by the smaller surface of contact. Why did the lateromedial orientation resulted in more frequent illusions with an ulnar deviation component (cf. Table 3), whereas the mediolateral condition, having the exact same surface of contact and position, caused less frequent ulnar or radial components?

**Table 3. Frequency of occurrence of different movements perceived depending on the motor's rotational orientation.**

| Movements perceived | Proximodistal rotation | Distoproximal rotation | Mediolateral rotation | Lateromedial rotation |
|---|---|---|---|---|
| **Uniplanar wrist movements** | | | | |
| Extension | 69.30% | 76.80%[a] | 70.80% | 57.70%[a] |
| Flexion | 2.40% | 1.60% | 1.50% | 2.30% |
| Ulnar deviation (UD) | 3.10% | 3.20% | 3.10% | 7.70% |
| Radial deviation (RD) | 3.10% | 0.00% | 2.30% | 1.50% |
| Total uniplanar | 77.9% | 81.6% | 77.7% | 69.2% |
| **Multi-planar/-joint movements** | | | | |
| Extension + UD | 3.10% | 4.00% | 2.30% | 7.70% |
| Flexion + UD | 0.00% | 0.00% | 0.00% | 2.30% |
| Flexion + RD | 0.00% | 0.80% | 0.00% | 0.00% |
| Extension + RD | 2.40% | 2.40% | 3.10% | 1.50% |
| Pronation | 6.30% | 0.00% | 6.20% | 5.40% |
| Supination | 5.50% | 4.00% | 4.60% | 7.70% |
| Circumduction | 0.80% | 0.80% | 0.00% | 1.50% |
| Total multi-planar/-joint | 18.10% | 12.00% | 16.20% | 26.10% |
| **No illusion** | 3.90% | 6.40% | 0.00% | 4.60% |
| **Total with extension component** | 74.80% | 83.20% | 76.20% | 66.90% |
| **Total with flexion component** | 2.40% | 2.40% | 1.50% | 4.60% |
| **Total with UD component** | 6.30% | 7.20% | 5.40% | 17.70%[b] |
| **Total with RD component** | 5.50% | 3.20% | 5.40% | 3.10%[b] |

[a]Denotes significant difference between conditions of rotational direction after Bonferroni correction (i.e., p ≤ 0.008).

[b]Denotes significant difference between directions of illusions perceived within a same condition of rotational direction after Bonferroni correction.

The only differential factor between the mediolateral and lateromedial conditions was the rotational direction of the mass (clockwise, counterclockwise). Shifting between clockwise and counterclockwise rotations likely changed. We propose that the resulting asymmetric elliptical orbits could change the angle and locus of maximal energy transfer between the vibrator and the participant in a way that favored the lateral side of the vibrator for the lateromedial condition, and the medial side for the mediolateral. Similarly, the distoproximal and proximodistal conditions would have resulted in a slight change in energy transmission favoring either the distal or the proximal part of the vibrator. From an anatomical standpoint, the effect of changing the rotational direction for the distoproximal and proximodistal conditions is negligible since both the proximal and distal sides of the vibrator are in contact with the same tendons. However, the situation differs for the mediolateral and lateromedial conditions. Indeed, the distal tendon of flexor carpi radialis is located slightly lateral to the wrist's center, near other flexor muscles (palmaris longus, flexor digitorum superficialis, flexor digitorium profundus). Therefore, placing the vibrator at the center of the palmar surface of the wrist like we did and using the lateromedial orientation (highest energy transfer toward the lateral side of the vibrator) would be more effective to target these wrist flexors and radial deviators. Our results did find that illusions with an ulnar deviation component (especially combined to extension) were significantly more frequent during lateromedial versus the other orientations, hence supporting the idea of a higher activation of flexor carpi radialis' muscle spindles. One would expect that the same applies to the mediolateral orientation and illusions with a radial deviation component (i.e., higher energy transfer on the medial side of the vibrator toward wrist flexors and ulnar deviators), however, the distal tendon of flexor carpi ulnaris is located farther from the center with its insertion near the fifth finger on the pisiform bone [31,32]. In line, we did not find higher occurrence of radial deviations when using the mediolateral or any other orientation. Still, perceptions of deviations and other complex multiplanar or multi-joint movements were sometimes perceived by our participants, possibly because vibration spread to nearby muscles (e.g., pronator quadratus). Of note, the higher variability of directions with the lateromedial orientation is probably not caused by more vague and weaker illusions that could have challenged the ability of the participants to identify the precise direction. Indeed, illusions were of similar clearness and speed and amplitude between all four orientations. This further supports that the lateromedial orientation is effective in vibrating tendons and creating kinesthetic illusions, but that this particular orientation preferentially stimulates different tendons (like the flexor carpi radialis) compared to the other orientations. Thus, our results support that changing the rotational direction of the eccentric mass and the vibrator's shape and surface of contact influence the effects of vibration, especially when targeting complex anatomical areas comprising several tendons unequally distributed. Although such results might only apply to vibrators of similar shape and design (i.e., eccentric rotating mass) than ours, they nonetheless highlight the importance of better considering the impact of these under-recognized technological and anatomical aspects of muscle tendon vibration. However, other factors unexplored by the present work should be considered in future studies, like the contribution of skin mechanoreceptors to kinesthetic illusions [33,34] since eccentric rotating mass vibrators also move back and forth parallel to the skin with a main direction of skin displacement likely shifting between the four tested conditions.

The fact that complex and variable illusions were perceived could be interesting for clinical applications, since multi-planar/joint movement are highly present in our daily lives when realizing functional tasks, such as reaching for an object [35]. Perceiving different movements without the need for changing the vibration's parameters and setup might lead to an enriched sensory experience and retraining. On the other hand, if a reproducible and direction-specific illusion is needed, for instance in research contexts, then our study recommends considering the vibrator's position and rotational orientation as factors to explore and control as much as possible. The simultaneous use of two or more vibrating motors around a joint could provide a better control of complex illusions and reduce variability between trials. Further research is needed to get a better grasp on all potential methodological factors that could influence vibration's efficacy to activate neurophysiological mechanisms of sensorimotor functions.

Our comparisons based on manual dominance shown less clear and lower speed and amplitude of illusions on the non-dominant side when the mediolateral rotation was used. Previous studies already highlighted handedness-related

differences of kinesthetic perception. Tidoni et al. found clearer movement illusion when vibration was applied to the non-dominant side, regardless of manual dominance. However, left-handed participants perceived better range of motion during illusory movement [23]. Another study found better ability to reproduce perceived movements on the non-dominant hand for right-handed men [36]. Hemispheric lateralization of sensorimotor functions and inter-hemispheric asymmetry could potentially explain differences of sensory perceptions between dominant and non-dominant sides. Studies using brain imagery suggested that the right hemisphere is more actively involved in conscious kinesthetic perceptions [30,37]. Since our sample was mostly constituted of right-handed individuals (n = 16/20), the relative contribution of hemispheric lateralization vs. dominance cannot be further untangled based on our data. Furthermore, the fact that between-side differences were only observed in one of our four conditions significantly reduce the strength of this finding. A bigger sample with an equal ratio of right and left-handed participants would be needed to answer this question properly.

## Conclusion

To conclude, our study raised awareness on new methodological factors to consider with the vibration-induced illusion paradigm, namely the position and orientation of the rotative mass relative to the targeted tendons. We stress out the importance of knowing the precise mechanical properties of the vibrator and the anatomical characteristics of the stimulated area, and to further identify and study other potentials factors of influence. On a clinical perspective, being able to generate complex and variable movement illusions with one vibrator like we found with the lateromedial orientation could be beneficial, for instance by recreating sensations more akin to everyday functional movements and keeping the patient engaged in the task. On the other hand, other orientations like the distoproximal should be preferred if more precise and reproducible illusions are required. Altogether, our study adds relevant knowledge to the literature and contributes to gaining an advanced understanding of the tendon vibration technique. Controlling all methodological-related factors of influence is critically needed before this technology can be implemented in clinical practice.

## Supporting information

**S1 File. Minimal dataset.**
(XLSX)

## Acknowledgments

The authors would like to thank all the participants who took part in our study.

## Author contributions

**Conceptualization:** Jacques Abboud, Francois Nougarou, Louis-David Beaulieu.

**Data curation:** Lydiane Lauzier.

**Formal analysis:** Lydiane Lauzier, Louis-David Beaulieu.

**Funding acquisition:** Jacques Abboud, Francois Nougarou, Louis-David Beaulieu.

**Methodology:** Jacques Abboud, Francois Nougarou, Louis-David Beaulieu.

**Project administration:** Louis-David Beaulieu.

**Resources:** Louis-David Beaulieu.

**Software:** Francois Nougarou.

**Supervision:** Jacques Abboud, Francois Nougarou, Louis-David Beaulieu.

**Visualization:** Jacques Abboud, Francois Nougarou, Louis-David Beaulieu.

**Writing – original draft:** Lydiane Lauzier, Louis-David Beaulieu.

**Writing – review & editing:** Lydiane Lauzier, Jacques Abboud, Francois Nougarou, Louis-David Beaulieu.

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
