## [Decision Letter · Decision Letter 0]

19 Aug 2024

Dear Dr. Beaulieu,

We look forward to receiving your revised manuscript.

Kind regards,

Monika Błaszczyszyn

Academic Editor

PLOS ONE

Journal Requirements:

"This study was support by Ordre Professionnel de la Physiothérapie du Québec (to L. Lauzier : https://oppq.qc.ca), Réseau Intersectoriel en Santé de l’Université du Québec (to J. Abboud, F. Nougarou and L-D. Beaulieu : https://risuq.uquebec.ca), FRQ Fonds de Recherche du Québec – Santé (FRQS) under Grand number 297854 (to L-D. Beaulieu : https://frq.gouv.qc.ca/sante/) and Réseau Provincial de Recherche en Adaptation-Réadapation (REPAR) (to L. Lauzier : https://repar.ca)."

6. Please include a separate caption for each figure in your manuscript.

7. Please include your tables as part of your main manuscript and remove the individual files. Please note that supplementary tables (should remain/ be uploaded) as separate ""supporting information"" files

Reviewers' comments:

Reviewer's Responses to Questions

**Comments to the Author**

1. Is the manuscript technically sound, and do the data support the conclusions?

Reviewer #1: Partly

Reviewer #2: Yes

2. Has the statistical analysis been performed appropriately and rigorously?

Reviewer #1: I Don't Know

Reviewer #2: I Don't Know

3. Have the authors made all data underlying the findings in their manuscript fully available?

Reviewer #1: No

Reviewer #2: Yes

4. Is the manuscript presented in an intelligible fashion and written in standard English?

Reviewer #1: Yes

Reviewer #2: Yes

Reviewer #1: It is interesting study to investigate the effect of rotational direction of vibration motor on kinesthetic illusion.

However, necessary details are missing for readers to understand the result.

First, it is hard to grasp the experimental procedure. Please add a proper figure describing the experimental procedure. Note that experimental procedure is critical for the kinesthetic illusion test, as the previous posture/movement of the upper limb critically affects the illusion of wrist joint.

Second, it is hard to understand the results because of the unfamiliar evaluation method (SKIP), which has not been well explained in the manuscript.

Although the SKIP has been introduced in authors' prior publication, authors should assume that most of the readers would not be familiar to that evaluation method.

Please add a figure describing the method (what each score means?). Also, please note that direction and amplitude should be clearly investigated, to describe the full aspects of the kinesthetic illusion. For example, if the illusion is "perfectly clear and precise", what that means in terms of angular illusion(how much differently the wrist angle was perceived)? If authors can provide any clue matching the strength of illusion and the angular illusion, that would be helpful for the readers to understand the impact of the intervention (vibration).

With these two information (exp procedure and exp result) well provided, I can fully review the manuscript.

Reviewer #2: The article is clear, well-structured, and thoughtfully organized, making it accessible to both specialists and non-specialists in the field. The authors have successfully presented their research in a logical sequence, ensuring that each section builds upon the previous one. Their adherence to both research and publication ethics is evident throughout the paper, as they have demonstrated transparency in their methodology and respect for ethical guidelines, particularly in data handling and participant confidentiality. The study's findings contribute meaningfully to the existing body of knowledge, offering new insights and potential directions for future research. Given the article’s clarity, ethical rigor, and valuable contribution, I find no areas that require revision or improvement and have no further comments.

**Do you want your identity to be public for this peer review?** For information about this choice, including consent withdrawal, please see our Privacy Policy

Reviewer #1: **Yes: ** Hangue Park

Reviewer #2: No

---

## [Author Response · Author response to Decision Letter 1]

4 Sep 2024

All responses to comments from Editor and Reviewers are in the 'Response to reviewers' file appended.

---

## [Decision Letter · Decision Letter 1]

30 Dec 2024

Dear Dr.  Beaulieu,

Thank you for submitting your manuscript to PLOS ONE. After careful consideration, we feel that it has merit but does not fully meet PLOS ONE’s publication criteria as it currently stands. Therefore, we invite you to submit a revised version of the manuscript that addresses the points raised during the review process.

We look forward to receiving your revised manuscript.

Kind regards,

Monika Błaszczyszyn

Academic Editor

PLOS ONE

Journal Requirements:

Reviewers' comments:

Reviewer's Responses to Questions

**Comments to the Author**

Reviewer #2: All comments have been addressed

Reviewer #3: (No Response)

2. Is the manuscript technically sound, and do the data support the conclusions?

Reviewer #2: Yes

Reviewer #3: Partly

3. Has the statistical analysis been performed appropriately and rigorously?

Reviewer #2: Yes

Reviewer #3: No

4. Have the authors made all data underlying the findings in their manuscript fully available?

Reviewer #2: Yes

Reviewer #3: Yes

5. Is the manuscript presented in an intelligible fashion and written in standard English?

Reviewer #2: Yes

Reviewer #3: Yes

Reviewer #2: The article is clear, well-structured, and thoughtfully organized, making it accessible to both specialists and non-specialists in the field. The authors have successfully presented their research in a logical sequence, ensuring that each section builds upon the previous one. Their adherence to both research and publication ethics is evident throughout the paper, as they have demonstrated transparency in their methodology and respect for ethical guidelines, particularly in data handling and participant confidentiality. The study's findings contribute meaningfully to the existing body of knowledge, offering new insights and potential directions for future research. Given the article’s clarity, ethical rigor, and valuable contribution, I find no areas that require revision or improvement and have no further comments.

Reviewer #3: (No Response)

**Do you want your identity to be public for this peer review?** For information about this choice, including consent withdrawal, please see our Privacy Policy

Reviewer #2: No

Reviewer #3: No

---

## [Author Response · Author response to Decision Letter 2]

14 Mar 2025

All comments are responded in a separate file.

---

## [Decision Letter · Decision Letter 2]

19 May 2025

Kinesthetic illusions induced by muscle tendon vibration: The orientation of the vibration motor as a new methodological factor.

PONE-D-23-38769R2

Dear Dr. Beaulieu,

We’re pleased to inform you that your manuscript has been judged scientifically suitable for publication and will be formally accepted for publication once it meets all outstanding technical requirements.

Kind regards,

Rasool Abedanzadeh, Ph.D

Academic Editor

PLOS ONE

Additional Editor Comments (optional):

Reviewers' comments:

Reviewer's Responses to Questions

**Comments to the Author**

Reviewer #3: (No Response)

2. Is the manuscript technically sound, and do the data support the conclusions?

Reviewer #3: Yes

3. Has the statistical analysis been performed appropriately and rigorously?

Reviewer #3: N/A

4. Have the authors made all data underlying the findings in their manuscript fully available?

Reviewer #3: Yes

5. Is the manuscript presented in an intelligible fashion and written in standard English?

Reviewer #3: Yes

Reviewer #3: I would like to thank the authors for taking my previous comments into account, which must have taken a lot of time and effort. I am aware that it is somewhat unconventional (and seemingly unfair) that the request for revision increased tremendously after the first round of revision, but I was not among the first-round reviewers who could point out the critical issues at the early stage. After the second round of revision, the manuscript is now much improved.

Nevertheless, I must stick a bit more with the point that vibration is independent of rotational direction. The authors refuted in terms of intuition, theory, and experiment, which is highly appreciated but not entirely convincing. To sum up, the intuitive explanation is wrong, and the theoretical explanation fails to support the authors’ claim (at least not by Refs 1 and 2); however, I cannot explain for the supporting evidence from the experimental measurements. Therefore, I believe it makes sense to present all results as they currently are, but the authors should make it clear that the reason for considering the rotational direction as a factor is because the accelerometer measurements suggest that the shaft orbit has a different shape (hence the vibration could be slightly different). Very importantly, this work is titled “The orientation of the vibration motor as a new methodological factor.” So even if there is a slight difference in vibration between the rotational directions, it is due to some internal properties of the motor itself and is irrelevant to the orientation. Rotational direction serves as a control factor, and should not be the focus of discussion.

As I will address later, a few revisions might be based on wrong interpretation of vibration dynamics. The authors may decide whether they should be corrected or not, but please keep in mind that this is basic physics, and if explained incorrectly, some readers will doubt the results of the main experiments.

Please see details in the attached PDF.

**Do you want your identity to be public for this peer review?** For information about this choice, including consent withdrawal, please see our Privacy Policy

Reviewer #3: No

---

## [Editor Report · Acceptance letter]

PONE-D-23-38769R2

PLOS ONE

Dear Dr. Beaulieu,

I'm pleased to inform you that your manuscript has been deemed suitable for publication in PLOS ONE. Congratulations! Your manuscript is now being handed over to our production team.

Kind regards,

on behalf of

Dr. Rasool Abedanzadeh

Academic Editor

PLOS ONE